# Relationship between PSD of Park Green Space and Attention Restoration in Dense Urban Areas

**DOI:** 10.3390/brainsci12060721

**Published:** 2022-05-31

**Authors:** Zhixian Zhu, Ahmad Hassan, Weijue Wang, Qibing Chen

**Affiliations:** 1School of Fine Arts and Design, Chengdu University, Chengdu 610059, China; aeo.12f.ctn@gmail.com; 2College of Landscape Architecture, Sichuan Agricultural University, Ya’an 625014, China; 3China Railway Eryuan Engineering Group Co., Ltd., Chengdu 610031, China; wwj901013@126.com

**Keywords:** dense urban area, park green space, perceived sensory dimension, attention restoration, moderator

## Abstract

Research shows that stress, a common problem in densely populated cities, can be relieved by exposure to the natural environment. As great significance has been attached to the relationship between the urban environment and public health, this paper aims to study the relationship and interaction between the perceived sensory dimensions of urban park green space, attention restoration, and state empathy. Therefore, we conducted an on-site questionnaire (PSD Scale) survey in four typical parks in Chengdu and recorded age, sex, daily stress, frequency of visits to parks, and other basic information from the respondents. In the survey and visit, we found that the group structure in the recreation area comprises chiefly of a few transient unfamiliar travelers and most long-haul neighborhood sightseers. Among long-haul vacationers, the greater part of them are moderately aged and older individuals in the encompassing local locations, whose lives are, for the most part, quick and proficient. Hence, to mirror the populace attributes of high-thickness metropolitan parks and to feature the agent bunches that have lived in the parks from here onward, indefinitely quite a while, the chosen bunches are somewhere in the range of 35 and 65 years old (half male and half female), so make sure there are no ailments, no drinking, and no late evenings in the earlier days, so a specific actual fundamental belief is kept up with. The main part of the exercise focused on the perceived dimension, state empathy, and attention restoration. The software SPSS24.0 was applied to the test of the validity and reliability of the perceived sensory dimension (PSD) Scale, and then the important correlation between the perceived sensory dimensions in the parks and visitors’ attention restoration was analyzed through multiple linear regression. Finally, the moderating effect of state empathy was tested by PROCESS. The findings show that (1) only seven dimensions in the PSD Scale are effective; (2) Serene and Refuge in the perceived sensory dimensions have a significant effect on the restorative components of attention. (3) Except for the dimensions of Rich in Species and Refuge, empathy enhanced the moderation effect in the interaction between the other five dimensions of the Perceived Restorative Scale (PRS), especially in the interaction between the Social and PRS dimensions. However, this topic needs to be further explored to provide a scientific basis and design strategy for research on the restoration potential of urban park green space in high-density urban areas.

## 1. Introduction

### 1.1. Need for Attention Restoration for Residents in Dense Urban Areas

#### 1.1.1. Social Stress Induced by Rapid Urbanization

Cities, as centers of human life, can provide not only better living conditions for people but also regional economic development. Urbanization has become a worldwide trend with a focus on the development of all countries [1]. Whatever views one holds about urbanization, they exert an increasingly subtle influence. As greater density is the essence of urbanization [2], some scholars state that high-density living is the future direction of urbanization and would change our way of life [3]. From the global perspective, there is an increasing trend for high density in the accelerating process of urbanization [4]. It is expected that in the next decade, urban populations will rise sharply in both developed and developing countries and may account for 85% and 56% of the total population, respectively [5]. The development of urbanization has placed increasing stress on city-dwellers [6]. Studies have found that the psychological stress experienced by the inhabitants is related to the negative feelings caused by crowded conditions [7]. Interpersonal proximity leads to more social interaction, a significant factor in generating psychological stress [8]. Lack of privacy caused by close distance causes anxiety [9]. The competition between space and social resources can also create psychological stress for the inhabitants [10].

#### 1.1.2. The Close Relationship between Stress and Attention Restoration

The Kaplans stated in their study that stress falls into two categories. One is a direct or perceived injury. The other is the lack of resources caused by intuition, chronic exhaustion, and the rise of autonomic arousal. The resources include external resources (such as money and social relations), physiological resources (such as physical strength and function), and psychological resources (such as attention and emotion) [11]. It can be seen that stress and attention restoration can be interactive and inter-transformational and that attention restoration can protect against stress [3].

Therefore, our brains need restoration to prevent stress [12]. Many scholars have discovered that the natural environment has the capacity to attract an innate affection from human beings who respond positively to the combination of features such as plants, water, stones, and animals [13]. The Attention Restoration Theory (ART) proposed by the Kaplans defines the four components of the restorative environment [14]. Being away (an environment that physically and psychologically detaches an individual from daily concerns and thoughts); Extent (immersion in an environment that is rich and coherent enough to constitute another world for exploration); Fascination (an environment that can hold one’s attention effortlessly); Compatibility (an environment matching the personal inclination and purposes). A green environment with these components can affect a gradual four-stage process of cognitive recovery from fatigue [15]. The first stage is a clearing of the mind and a restoration of peace of mind. The second is voluntary attention to fatigue recovery. The third stage allows the individual to be gently distracted and engaged in a low-stimulation activity, which reduces the internal noise and provides a quiet internal space for relaxation. The fourth stage, also the deepest stage, is a reflection on important personal matters, such as priorities, actions, or goals.

### 1.2. Related Concepts of Park Green Space in Dense Urban Areas

#### 1.2.1. Park Green Space

Around the world, scholars and institutions may differ in their definition of park green space (PGS). In his definition of PGS, Olmsted, the architect of New York’s Central Park, focuses on its green and public attributes and the functions it serves, i.e., the functional public green space apart from the gray areas (mainly artificial structures such as buildings, roads, squares, and facilities). However, the Swedish architect, Blom, defines PGS as a natural and cultural complex reconstructed on the basis of the existing natural environment and lays stress on its ecological and cultural elements. In his book “Landscaping,” Taiwanese scholar Lin Lejian defines PGS as a green space where (1) the public can enjoy activities for joy, leisure, relaxation, and recreation in order to keep themselves healthy and raise their cultural awareness; (2) the public can have free access to its supporting facilities; (3) the public may find refuge against natural disasters. In recent years, China has issued several standards related to park green space (PGS), which are described below. (1) There is the Standard of Basic Terms for Urban Planning (GB/T 50280-98), in which the green space is defined as a designated space for ecological improvement, environmental protection, recreation for the residents, and landscaping. The Standard classifies the green space into the following five categories: park green space, production green space, protection green space, supporting green space, and other green space. Park green space refers to the public green space open to the public for recreation, which should be of adequate size and have a substantial green area and service facilities. (2) There is also the Standard of Classification of Urban Green Space (CJJ85-2002T) and the Standard of Urban Land Classification and Construction Land Planning (GB50137-2011), both of which define PGS as the public green space open to the public for recreation with the function of ecology, beautification, and emergency. This paper takes the Standard of Urban Land Classification and Construction Land Planning (GB50137-2011) as the definition of park green space for research and classification.

#### 1.2.2. Dense Urban Area

The term “dense urban area” generally refers to a limited urban area with a high floor-area ratio, high population density, high building coverage, highly concentrated high-rise buildings, and low space openness. It serves the following functions of a city: complex and compatible, compact and intensive. Academically, there is no agreed quantitative criterion as to what a dense urban area is, but it is generally agreed that it involves both building density and population density. A city or urban area with a population of 15,000 and over p/km^2^ and a 2.0 and over floor-area ratio can be regarded as a dense urban area [16,17]. It can be seen that park green space in dense urban areas has the following distinctive characteristics: limited area, a variety of functions, high usage, greater openness, compatibility with the surroundings, and a variety of users.

#### 1.2.3. Park Green Space in Dense Urban Areas

Based on the two concepts above, parks in dense urban areas have to strike a balance between limited land resources and functions and integrate the public space and the natural environment to meet the needs of recreation and health [18].

Chengdu is a highly developed city in Sichuan Province, China, with a high population density. In 2017, Chengdu had a population of 6.57 million in a 371 km^2^ urban area, exceeding the criterion of 15,000 people/km^2^, while in the central urban area (about 60 km^2^, within the 2nd Ring Road), the population density has gone far beyond the criterion. As a result of the circle-layer spatial structure created during the process of urbanization, the park green spaces in this area are mainly distributed in spots and belts. Based on what has been mentioned above, the central urban area of Chengdu is an ideal area for research on the functions of the park green space in a dense urban area. This study chooses the dense urban area—the central urban area within the 2nd Ring Road—as the area for research. Here, four representative free public parks have been selected as the sample plots for the experiment, namely, Wangjiang Tower Park, Southern Suburban Park, Huanhuaxi Park, and People’s Park.

### 1.3. Moderation Effect of Empathy on Environment Perception and Psychology

Perception involves both simple senses and complicated awareness [19]. In practice, the senses and awareness are virtually inseparable, for they are part of a continuous process in which awareness reflects the sensory stimuli and converts them into an organized experience [20]. Awareness further processes the information derived from the senses. If you watch the leaves falling from the trees in a park in the autumn, you are receiving sensory information. However, if you feel lonely and melancholy at the sight of the falling leaves, it is your awareness that is at work. Rudolf Arnheim believes that vision is the mechanical perception of objects, while visual awareness is the perception of the expressivity of objects, a particular aesthetic awareness [21]. The perception of landscapes can be regarded as a process in which senses and awareness are interwoven [22]. The experiencers have similar senses about the same object, but they may differ greatly in awareness. Generally, the senses are within the scope of psychological study [23], while awareness is within the orbit of philosophical study. Based on phenomenology, Merleau-Ponty believes that the continuous movement of one’s body in a space integrates the subjective world and the objective world. In this process, awareness associates symbols with meanings and forms with content. Hence, the perception of the landscape derives from both physical and psychological cognition, and the awareness of the landscape synthesizes all the senses and converts them into a complicated and continuous experience of landscape perception.

In all these perception experiences, vision plays a leading role. People rely more on vision than the other senses, as 87% of external information comes through this channel [24]. Sensory stimuli, to a great extent, govern one’s perception of the environment.

Sobel believes that empathy is the basis for studying perceived behaviors related to the environment [25]. Researchers have found that empathy is a neural representation [26], and the ultimate goal of the reactivation of the neural presentation is to help an individual perceive the environment and the connotations and spirit it carries and produce similar or the same behaviors and emotions. Preston and de Waal presented their Perception-Action Model (PAM), believing that shared representation is the basis of empathy. When an individual perceives the emotions of others, a shared representation will be activated, and one will experience similar emotions to others [27]. Cognitive empathy is explained by psychological theories.

Brain imaging has shown that psychological theories about empathy mainly involve brain areas such as the sulcus temporalis superior (STS), temporoparietal junction (TPJ), temporal pole (TP), and medial prefrontal cortex (MPFC), which form a neural network that can represent other people and self. Among them, the MPFC plays a significant role in understanding other people’s psychology and cognition of the surrounding environment. Some scholars have shown the respondents figures related to psychological theories and empathy. Their findings have revealed that MPFC and TPJ are active in the processing of emotional information and speculation about the psychological inferences of other people [28]. Both activations would, to a certain extent, relieve anxiety, fear, and psychological stress [29]. In this process, the brain has been enhanced and drilled through deliberate exercises. The efforts of the subjective will bring about psychological change, i.e., neuroplasticity. By “planting” certain sentiments and deliberately adjusting one’s cognition, perception, and behaviors, a person can improve his/her related brain areas and psychological state. Empathy on exposure to the external environment will generate greater sensitivity in the cerebral activities, representing an adaptive advantage [29]. It suggests a certain but not an absolute link between empathy and psychology. Other correlations still remain to be researched.

It can be seen that empathy exerts a certain moderating effect on environmental perception and psychology. The correlations between the three can be established on the basis of statistical moderation analysis, the results of which can be found in the following sections.

### 1.4. Research Design

Figure 1 shows that by introducing control variables, the study examines the mechanism of the effect of independent variables and moderator variables on dependent variables [30]. Various types of individuals experience various qualities in a similar climate, some of which are useful to human wellbeing [31]. Estimation of natural ascribes according to the perspective of buyer discernment can demonstrate highlights that cannot be distinguished by genuine estimation [32]. Besides, the distinguished elements, according to a perceptual perspective, are straightforwardly connected with reclamation, which assists scientists with figuring out which biological highlights are well known or restored. There is an association between individuals’ impression of the climate through their faculties and individuals’ well-being [33]. The PSDs were created to look at the attributes of the perceptual helpful climate, which comprises of the following eight aspects: nature (wild nature not made by people), culture (a climate wherein the pith of human culture is available). Prospect (a region with an open view), social (a climate that is prepared for social exercises), space (a green climate that is roomy and free and has a specific measure of lucidness in it), loaded with species (A climate that offers an assortment of creatures and plants), Shelter (a covered up and safe climate where individuals can see others moving), Seren (a tranquil and safe spot) [34]. Specifically, PSDs are custom-made to individuals’ requirements for unwinding, working out, social association, diversion, and security, so they can recognize natural elements that decrease pressure and advance human recovery [35], subsequently offering bearing for building a climate with great pressure rebuilding. The PSDs have been perceived as a successful instrument for normalized examination and assessment of the seen highlights of the rebuilding climate, which assists with estimating the nature of green space for restoration tests [36]. There are likewise suggestions for their great application in China’s social and natural settings [30].

The connection between PSDs and stress restoration presented through natural recovery experience has forever been an interesting issue. In the examination of the survey, with the rebuilding of strain in metropolitan parks, nature, wealthy in species, peacefulness, and asylum [37], while little open metropolitan green space (SPUGS), social and serenity were fundamental [38]. The consequences of research center analyses recommended that every one of the eight PSDs fundamentally affected pressure recuperation and that nature and serenity were the main variables [39]. Quiet, outcast, and nature were viewed as related to pressure restoration for undergrads [40]. Additionally, in young adult investigations, the impact of reclamation of weight on nature, asylum, and potential was found [41]. For patients with stress-related ailments, an eating routine wealthy in shelters, serenity, nature, and plantation species can influence pressure recuperation [42]. It very well may be seen that on the off chance that, the examination strategies, kinds of climate, and subjects are different, the exploration aftereffects of PSDs will be unique. Furthermore, further examinations are expected to investigate the connection between PSDs and stress restoration. Moreover, with regards to Chinese culture and natural foundation, it is not clear what PSDs have to do with stress restoration. Based on the analysis of the existing relevant literature, the research mainly focuses on the effects of perceived sensory dimensions (PSDs) on attention restoration and the moderation effect of the perceived sensory dimensions on attention restoration.

The paper proposes the following hypothesis:(1)The perceived sensory dimension (PSD) scale has a certain degree of reliability and validity;(2)Eight perceived dimensions have different effects on components of attention restoration;(3)The relationship between eight perceived sensory dimensions and attention restoration can be moderated by empathy.

## 2. Materials and Methodology

The experiment was conducted in April when the weather was sunny. Questionnaires (PSD Scale) [43] were distributed to visitors in the four parks in the mornings, afternoons, and evenings of the weekdays and all weekends. In the questionnaire, subjects are as follows: Wangjianglou Park is mainly 45–65 years old, Huanhuaxi Park is 35–60 years old, People’s Park is mainly 55–65 years old, Nanhu Park is mainly 35–40 years old and 60–65 years old (Figure 2). In the four parks, 75% of the subjects were surrounding residents, including teachers, engineers, company employees, park management workers, retired employees, and so on. The frequency of visits to the park is 3–4 times a week, with non-retirees visiting the park mainly on weekday evenings and weekends. Although there were no more than 100 people at each test site, they lived in high-density enclosures and were therefore representative.

As one of the most evolved urban areas in Sichuan Province and, surprisingly, the southwestern locale, Chengdu has an exceptionally high populace thickness. As indicated by the review, toward the finish of 2017, the number of inhabitants in the “focal metropolitan region” of the city was roughly 6.57 million, with a developed area of around 371 km^2^ and a populace thickness of around 17,700 people/km^2^, which is the breaking point for high-thickness urban areas. Quality (15,000 people/km^2^). Deeply and midsection of the focal metropolitan region, the populace thickness of the “Downtown Area of Chengdu” (covering an area of roughly 60 km^2^, which is a metropolitan constructed region inside the second ring street of the old city). Has surpassed 15,000 individual’s standard/km^2^. What is more, to concentrate on the medical advantages of Park Greenspace with regard to high-thickness metropolitan regions, the “exposure” of Park Greenspace should be completely thought of, and the recreation area’s “huge base” and “Diverse sorts “tests ought to be featured. Consequently, in choosing to concentrate on areas, Park Greenspace which can oblige huge help limit, free open, can oblige an enormous number of examples and species, and extensive capacities ought to be chosen. The four test points of this article have such elements.

This exercise aimed to capture the immediate experience of the visitors [44]. The scale consisted of three parts. The first part included personal data, such as age, sex, occupation, stress self-assessment and frequency of access to parks, and comfort index of the body. The comfort index of the human body involves three environmental indexes, namely, air temperature, humidity, and wind velocity. Its calculation method is based on the following formula:SSD = (1.818t + 18.18)(0.88 + 0.002f) + (t − 32)/(45 − t) − 3.2v + 18.2

In the above equation, SSD represents the comfort index of the human body, t is the average temperature (°C), f is relative humidity (%), and v is wind velocity (m/s).

The second part is concerned with qualitative analysis of the characteristics of the urban park green space based on PSDs, based on 1000 randomly selected questionnaires, defining the following eight dimensions: Serene, Space, Nature, Rich in Species, Refuge, Culture, Prospect, and Social. Karin Kragsig Peschardt and Urika Karsson Stigsdotter from Denmark proposed a method for estimating the distinction in open inclinations for green spaces through little open green space reclamation impact and discernment rebuilding aspects. The components of mental rebuilding incorporate the following accompanying eight classes: Nature, Culture, Possibility, Social, Space, and Species of Species, Species, Refuge, and Serene, every one of which is decided by various elements. In the trial, they chose nine little open green spaces in Copenhagen’s high-thickness regions and positioned them as indicated by eight reasonable aspects. Two expert scene designers investigated the components of the green space idea in each model. Then, at that point, members’ indicative information on the natural restoration of nine examples was obtained through a field poll. The survey utilized incorporated a “need” perspective in view of PRS, with an aggregate of 24 articulations.

Through the score of each greenspace test that compares to the element of every insight and through the measurements of the PRS scale information, a few informational collections were obtained as follows: The view of the green space in each example. An aspect rating table, posting the five-layered typical score of “similarity”, “Consistency”, “Distance”, “Appealing”, and “Need” are the green spaces of each example, and a contact rundown of the five aspects and perceptual aspects acquired through relapse examination. An examination of the rundowns shows that the assessment of every PRS aspect is affected by various PSD aspects, and some PSDs additionally assume a conclusive part in natural strength.

From one perspective, the PSD perception dimension model looks at residents’ topical sentiments towards the climate. Then again, it likewise acquires an expert appraisal of the green space climate through proficient evaluations. Simultaneously, the perception recovery scale (PRS) and the perception recovery dimension (PSD) have been consolidated to exhibit the capacity of dynamic recuperation through strong natural components, as a source of perspective for future public green space plans, can be utilized.

In addition, degrees of empathy were assessed based on Batson’s Emotional Response Scale (ERS), [45] which measures the immediate emotional response to a specific context, the degree of empathy through state empathy, and effective control of state empathy. The third part is about the assessment of the attention restoration of the respondents through the perceived restorative scale (PRS). The PRS was first introduced by Hartig, Korpela, Evans, and Gärling in 1996 [46] and based on the four components of the restorative environment. Currently, there are several versions. The perceived restorative scale (PRS) in this paper is the Chinese version introduced by Ye Liuhong and Wu Jianping in 2010. It includes 22 items on a 7-point scale (1 = not at all, 7 = completely). It took the visitors about six minutes to finish the questionnaire. Finally, 87 validated questionnaires were collected in Group Wangjiang Tower Park, with a 96.67% validation rate, 90 questionnaires in Group Southern Suburban Park, with a 100% validation rate, 85 questionnaires in Group Huanhuaxi Park, with a 94.44% validation rate, and 86 questionnaires in Group People’s Park, with a 95.56% validation rate. The total number is 348. In the process of sorting out the questionnaires, we found that a small part of the questionnaires were not carefully filled in by the subjects, relevant options were omitted, or the selection rate was too high, so the questionnaires were regarded as invalid and should not be used in the statistics. Some of the questionnaires are shown in Figure 3.

## 3. Results

### 3.1. Assessment of Reliability and Validity of PSD Scale

The reliability and validity of the PSD Scale were tested before the research analysis. The reliability was tested using Cronbach’s α coefficient and composite reliability, while validity was tested by factor analysis and average variance extracted (AVE).

#### 3.1.1. Assessment of Reliability

The test of reliability is to describe the expression level of the observed variables to potential variables to assess the consistency of the PSD Scale and its reliability. It has been academically agreed that a Cronbach’s α coefficient over 0.8 is relatively reliable [47]. We tested the statistical reliability of the PSD Scales. Their Cronbach’s α coefficients are all over 0.8, indicating a high internal consistency and reliability (Table 1).

#### 3.1.2. Assessment of Validity

This paper conducted the confirmatory factor analysis (CFA) based on exploratory factor analysis (EFA) to determine the structural relationship between the factors and each dimension. The validity of the Scale was assessed with a combination of the composite reliability (CR) and the average variance extracted (AVE).

##### Analysis of EFA

Thirty-five statements in the PSD Scale went through the Kaiser–Meyer–Olkin (KMO) test and Bartlett test to test their suitability as factors. According to the academic requirement, a KMO value of over 0.7 indicates the adequacy of sample data for factor analysis [48]. As is shown in Table 2, the values of the two groups were both over 0.7, while the statistical *p*-value of the Bartlett X2 test was 0.000, <0.01, indicating that the sample data were correlated and concentrated, and suitable for factor analysis (Table 2).

##### Confirmatory Factor Analysis (CFA)

The CFA includes both convergent and discriminant validity assessments. Convergent validity assessment involves calculating factor loadings, AVE (average variance extracted), and CR (composite reliability). The CR values are desirable when the data are greater than the threshold values 0.5, 0.45, and 0.7, respectively, while the discriminant validity value assessment mainly depends on √AVE value. If it is greater than the correlation coefficients between each dimension, it meets the requirement of discriminant validity. (1) In terms of factor loadings, estimate values of all items in the four parks are greater than 0.5, indicating their high representation in all the parks. (2) AVE and CR values show that AVE values, with the exception of that of 0.363 in Wangjiang Tower Group in the dimension Nature, are all greater than the threshold value of 0.45, indicating that the convergent validity of the PSD in all parks, except the dimension Nature in Wangjiang Tower Park, is desirable. (3) In the assessment of discriminant validity, √AVE values of eight dimensions in each park are all greater than the correlation coefficients between each dimension, indicating a desirable discriminant validity in PSD.

In conclusion, the assessment of the reliability and validity of PSD finds that the convergent validity assessment and the critical values in the dimension Nature are not desirable (there are no significant differences in other dimensions). It will be omitted in the dimension Nature of PSD to guarantee its rigor and scientificity. For the sake of simplicity, the other seven perception dimensions will be integrated into a unit of seven groups of perception dimensions for convenience, making it possible to analyze them integrally with other variables. This approach guarantees efficiency in analysis, and the increase in the samples improves the accuracy of the analysis.

### 3.2. Regression Analysis of Interaction between Each PSD and the Restorative Components

#### 3.2.1. Assessment of Collinearity

As multiple regression analysis involves collinearity, it is necessary to test the collinearity of the independent variables. If the independent variables are highly correlated, the regression equation is unlikely to be stable. Tolerance and VIF (variance inflation factor) are usually applied to the multiple statistical regression analysis between the independent variables. VIF is defined as the reciprocal of tolerance. When both VIF and tolerance are close to 1.00, there will not be significant collinearity between the independent variables. Generally, a VIF < 10 is acceptable in multiple regression analysis [49].

In this paper, VIF values of the constants such as sex, age, and education are around 1.000. The VIF values in seven perceived sensory dimensions (PSDs), Culture, Prospect, Social, Space, Rich in Species, and Serene, are, respectively, 2.253, 2.615, 2.238, 3.747, 3.753, 5.037, and 2.463, indicating regular collinearity, and it is feasible to conduct a regression analysis of the interaction between the PSDs and physiological and psychological indexes.

#### 3.2.2. Regression Analysis

Based on the regression analysis of each perceived sensory dimension on Being Away, Fascination, Extent, and Compatibility, the R-squared is above 10%, and in the components Being Away and Fascination, it is above 30%, indicating that eight PSDs can interpret the 10% variance of the attention restoration, each dimension contributing uniquely to attention restoration [50]. The F-test also shows that explanatory variables have a significant effect on dependent variables. Furthermore, the Durbin-Watson (DW) statistic shows a value close to 2 in the four components [49]. It can be concluded from the above that the regression analysis of the interaction between the eight dimensions and the four components of attention restoration is effective (Table 3).

Observation of R-squared from Table 4 finds that state empathy has some moderation effect on the interaction between all PSDs and attention restoration except dimension Rich in Species. In addition, the F-test shows that the moderation effect is statistically significant, which statistically supports the moderation effect on the interaction between the perceived dimensions and attention restoration. The PRS is significantly reliable and valid, including the following four components: Being Away, Fascination, Compatibility, and Extent, which can interpret 57.05% of the population variance. The correlation coefficient between the four subscales and the total scale is between 0.724 and 0.943, and the coefficient between the subscales is between 0.478 and 0.684. The Cronbach α coefficient of the total scale and four subscales is between 0.769 and 0.936, and their split-half reliability is between 0.695 and 0.903.

## 4. Discussion

### 4.1. The Relation between Each Perceived Sensory Dimension and Components of Attention Restoration

Previous studies have emphasized that the green environment has a restorative effect [51]. Scholars such as Cooper-Marcus (1998) [52], St Leger (2003) [53], and Grahn P (2010) [54] have found that immersion in nature can effectively relieve fatigue, anger, and worry. Continuous attention and interest can concentrate the attention and increase the sense of joy. These findings have supported the statement that the PSDs have some effect on attention restoration. It can be seen from Table 3 that some of the dimensions and the attention restoration are significantly correlated. (1) As to the restorative component Being Away, the dimension with the most predictive effect is Serene, with significance *p*-values of 0.000, all less than 0.01, followed by the dimensions Refuge and Space, with respective significance *p*-values of 0.011 and 0.046, all less than 0.05. This finding indicates that Serene can help people get rid of daily stress and obligations more effectively than other dimensions, a finding that has also been reflected in other studies on the relationship between PSDs of small urban green space and attention restoration [55]. Besides, dimension Refuge and Space can also help people temporarily forget the realities of daily life. (2) As for the restorative component Fascination, three dimensions, Rich in Species, Refuge, and Serene, have greater predictive effects, with respective significance *p*-values of 0.001, 0.00, and 0.001, all less than 0.01, indicating that these three dimensions can hold a person’s attention and stimulate curiosity. (3) As for the restorative component Extent, the dimensions Social and Refuge have the most predictive effect, with respective significance *p*-values of 0.003 and 0.009, both less than 0.01, indicating that the two dimensions help improve people’s cognition of the environmental varieties. (4) As for Compatibility, the dimension Serene is the dimension with the most predictive effect, with a *p*-value of 0.009, less than 0.01, indicating that the Serene can generate a sense of belonging and create a state of harmony between a human being and nature.

It can be concluded that the dimension Serene is significantly related to the following three restorative components: Being Away, Fascination, and Compatibility. Refuge, on the other hand, is related to Being Away, Fascination, and Extent, indicating that Serene and Refuge are the keys to perceived sensory dimensions that influence attention restoration. The previous findings, however, show that Serene and Nature share a similar predicting effect [55], which is a little different from our study. It may be that the dense urban park green spaces have a high frequency of access, in contrast to other green spaces. People may not feel Nature requires Serene to match in an environment and is no longer an important way to relieve stress in a situation where safety, inclusivity, recreation, and capacity of an environment are the major influential factors for attention restoration. Besides, Space, Rich in Species, and Social are also important, influential dimensions for attention restoration, but with single and limited influential levels. For example, the dimension Space is only related to the component Being Away, Rich in Species only to Fascination, and Social only to Extent.

### 4.2. Moderation Effect of State Empathy on the Interaction between Each PSD and Attention Restoration

It can be concluded from Table 4 that R-squared in Rich in Species is not significant, indicating empathy cannot moderate effectively between Rich in Species and psychological health. It shows that r^2^ in species richness is not significant, indicating that empathy cannot regulate the relationship between species richness and mental health. With a simple slope analysis as in Figure 4 below, the following is applied to all the dimensions except Rich in Species. It means the degree of regulation was further analyzed under the condition that the regulation variables had a significant regulation effect, so species richness was excluded.
(1)Respondents with a higher empathy degree (M + 1SD) showed a significant predictive effect between dimension Culture and attention restoration, simple slope = 3.956, t = 3.787, *p* < 0.01, while respondents with a lower empathy degree (M − 1SD) also showed a significant predictive effect, simple slope = 3.607, t = 2.137, *p* < 0.05. Besides, simple slope = 3.956 > 3.607. This suggests that those with a higher empathy degree show a slightly greater moderating effect than those with a lower empathy degree between dimension Culture and attention restoration but without a highly significant difference;(2)Respondents with a higher empathy degree (M + 1SD) showed a significant predictive effect between dimension Prospect and attention restoration, simple slope = 3.600, t = 6.157, *p* < 0.01, while respondents with a lower empathy degree (M − 1SD) showed a significant predictive effect, simple slope = 3.327, t = 3.264, *p* < 0.01. Besides, simple slope = 3.600 > 3.327. This finding suggests that those with a higher empathy degree show a slightly greater moderating effect than those with a lower empathy degree between dimension Prospect and attention restoration but without a highly significant difference;(3)Respondents with a higher empathy degree (M + 1SD) showed a significant predictive effect between dimension Social and attention restoration, simple slope = 3.321, t = 7.164, *p* < 0.01, while respondents with a lower empathy degree (M − 1SD) also showed a significant predictive effect, simple slope = 2.819, t = 2.121, *p* < 0.05, *p* < 0.05. Besides, simple slope = 3.321 > 2.819. This finding suggests that those with a higher empathy degree show a slightly greater moderating effect than those with a lower empathy degree between dimension Social and attention restoration but without a highly significant difference;(4)Respondents with a higher empathy degree (M + 1SD) showed a significant predictive effect between dimension Space and attention restoration, simple slope = 3.055, t = 7.5146, *p* < 0.01, while respondents with a lower empathy degree (M − 1SD) also showed a significant predictive effect, simple slope = 2.785, t = 3.749, *p* < 0.01. Besides, simple slope = 3.055 > 2.785. This finding suggests that those with a higher empathy degree showed a slightly greater moderating effect than those with a lower empathy degree between dimension Space and attention restoration but without a highly significant difference;(5)Respondents with a higher empathy degree (M + 1SD) showed a significant predictive effect between dimension Refuge and attention restoration, simple slope = 1.307, t = 6.210, *p* < 0.01 while respondents with a lower empathy degree (M − 1SD) did not show a predictive effect between the dimension Refuge and attention restoration, simple slope = 1.145, t = 1.401, p > 0.05;(6)Respondents with a higher empathy degree (M + 1SD) showed a significant predictive effect between the dimension Serene and attention restoration, simple slope = 3.626, t = 5.162, *p* < 0.01, while respondents with a lower empathy degree (M − 1SD) also showed a significant predictive effect, simple slope = 3.378, t = 3.325, *p* < 0.01. Besides, simple slope = 3.626 > 3.378. This finding suggests that those with a higher empathy degree show a slightly greater moderating effect than those with a lower empathy degree between dimension Serene and attention restoration but without a highly significant difference.

### 4.3. The Optimal Restorative Dimensions and Their Combination in Urban Park Green Space

#### 4.3.1. Transformation of Single Dimensions

The interaction between perceived sensory dimensions and attention restoration shows that the best dimensions in the urban park green space are Serene and Refuge. As for the dimension Serene, improvement of its moderating effect means controlling the noise and ensuring sanitary conditions in the surroundings. For example, we may make full use of the plants or landscape elements to prevent noise, thereby improving the Serene effect. At the same time, natural sounds, such as those from the wind, water, and animals, and cultural sounds, such as the music of a drum or the chimes from a bell tower, may also promote the moderating effect. As for the dimension Refuge, factors such as security and comfort of the facilities and space for activities must receive priority. A space that is likely to lead people to get lost should be avoided. The prevention of potential dangers also should be considered in the design of the railings, materials choice, contact with water, and plant configuration. The number of visitors and vehicles should be strictly controlled so they do not exceed the capacity limit and create crowdedness.

#### 4.3.2. Innovation in Combination of Different Dimensions

When empathy is taken as a variable, it has been observed that visitors with a higher degree of empathy can more easily achieve attention restoration in the dimensions Culture, Prospect, Space, and Serene, particularly in the dimension Social. Through the analysis above, the paper proposes the following possible optimal design: (Culture + Prospect + Space + Serene) × Social.

(1) Singular Interactive Mode

Singular Interactive Mode refers to the following five modes: Culture × Social, Prospect × Social, Space × Social, and Serene × Social. In the mode Culture × Social, the cultural elements, through direct application, extension, or artistic conception, primarily through horizontal relevance and vertical extension of these elements, can be applied to the improvement of cultural transmission through the construction of a multilayer, multiangle, comprehensive, and compatible interactive cultural platform, which can be a harmonious anchor for the residents by representation over abstraction. The mode Prospect × Social may focus on the accessibility and relevance of accessible facilities such as the greenery, the water system, and the traffic system. Other considerations may include maintenance service and management and capacity control. The mode Space × Social may serve as guidance for social behaviors under different forms of space. For example, static activity spaces may guide people to reduce physical and verbal interaction and generate greater visual, auditory, and olfactory perception. However, dynamic activity spaces may focus on greater verbal and physical interaction to sustain people’s emotional interactions. As for the mode Serene× Social, some advanced technology may be introduced to the soundscapes to break the conventional design, engendering interaction, and achieving landscape-person interaction. 

(2) Integrated Interactive Mode

Integrated Interactive Mode refers to a variety of dimensions interacting with the dimension Social, such as (Culture + Prospect) × Social, (Serene + Space) × Social, (Culture + Prospect) × Social), (Culture + Prospect + Serene) × Social, (Serene + Space + Prospect) × Social, or (Culture + Prospect + Space) × Social, or any other combinations. The design may refer to the design of the singular interactive mode. It can be concluded that the design of the park green space in dense urban areas should meet a greater variety of needs and make full use of the limited green resources to achieve the optimum physical and mental restoration effects.

## 5. Conclusions

Concerning park green space in dense urban settings, this paper, based on previous studies, proposes for the first time a model of the relationship between perceived sensory dimensions (PSDs) and perceived restorative scale (PRS), introduces empathy as a moderator variable, and further explores the interaction between the three aspects. The findings show the following:(1)The PSD Scale test showed that all the dimensions except Nature had relatively high reliability and validity, meaning PSD can be applied as a measuring tool for the perceived sensory dimensions in the urban park green space;(2)Through multiple linear regression analysis, the paper explores the effect of different perceived sensory dimensions on the components of attention restoration. The findings show that PSDs can to some extent interpret attention restoration, suggesting that PSD is an important predictive variable for this purpose. Generally, the predictive effect lies in the following three dimensions: Refuge, Serene, and Rich in Species. In terms of components, the Dimensions Space, Refuge, and Serene exert stronger predictive effects on the component Being Away of PRS; the Dimension Rich in Species exerts a stronger predictive effect on the component Fascination of PRS; the Dimensions Social and Refuge, exert stronger predictive effects on the component Extent of PRS; the Dimension Serene exerts stronger predictive effects on the component Compatibility of PRS;(3)Hierarchical regression analysis was used to assess the moderating effects of the degree of empathy between PSDs and attention restoration. The findings show that the degree of empathy does not show a significant moderating effect between Rich in Species and attention restoration. Further simple slope analysis through PROCESS plugins finds that higher degrees of state empathy show significant moderating effects between dimension Refuge and attention restoration, and lower empathy degrees do not show significant moderating effects. In the other five dimensions, both higher empathy degrees and lower empathy degrees show significant moderating effects. Furthermore, empathy degrees of both types had enhanced moderating effects between the Culture, Prospect, Social, Space, and Serene five dimensions and attention restoration, particularly in the dimension Social, without significant difference.

In conclusion, the proposed hypotheses in this paper have passed model validation. There are, however, some limitations in the process that need further exploration and improvement. First, although the PSD Scale has been based on previous studies, with good reliability and validity, and tested through questionnaires, there may be room for improvement. The PSD Scale should be redesigned to improve its validity. In addition, part of the content in the scale is a little hard to understand, so most of the valid samples are young people, suggesting the samples may not represent all age groups. Further research will comparatively analyze the PSDs in different groups, leading to a more reliable and valid finding. Finally, the relation between PSDs and attention restoration may be influenced by more moderator variables, even intermediate variables, than just empathy, which is the moderator variable in this paper. All these points could be addressed in further research.

## Figures and Tables

**Figure 1 brainsci-12-00721-f001:**
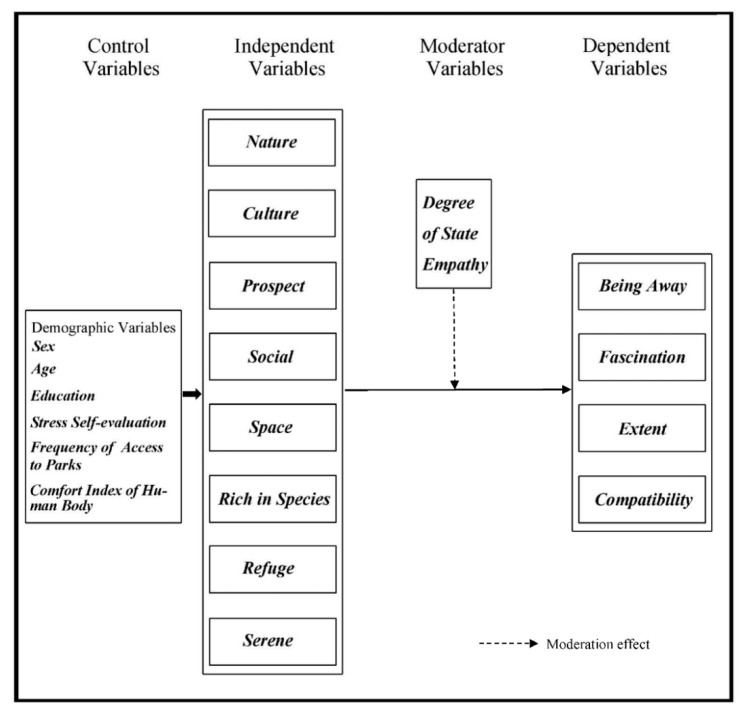
Theoretical Model for Perceived Sensory Dimensions and Attention Restoration in Urban Park Green Space.

**Figure 2 brainsci-12-00721-f002:**
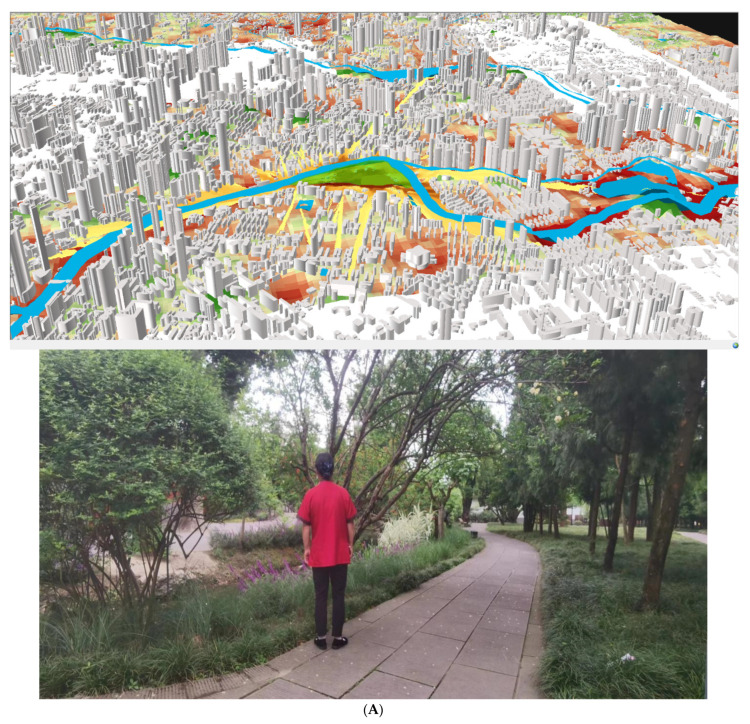
(**A**) Perspective view of Wangjiang Building Park; (**B**) perspective view of Nanjiao Park; (**C**) perspective view of Huanhuaxi Park; (**D**) perspective view of People’s Park.

**Figure 3 brainsci-12-00721-f003:**
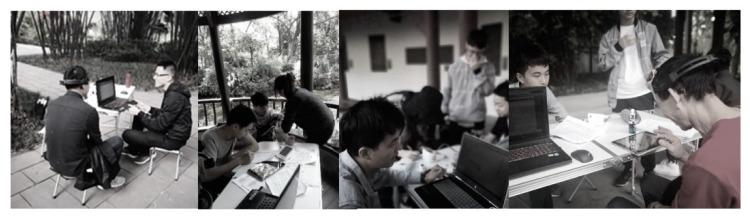
On-the-spot experiment.

**Figure 4 brainsci-12-00721-f004:**
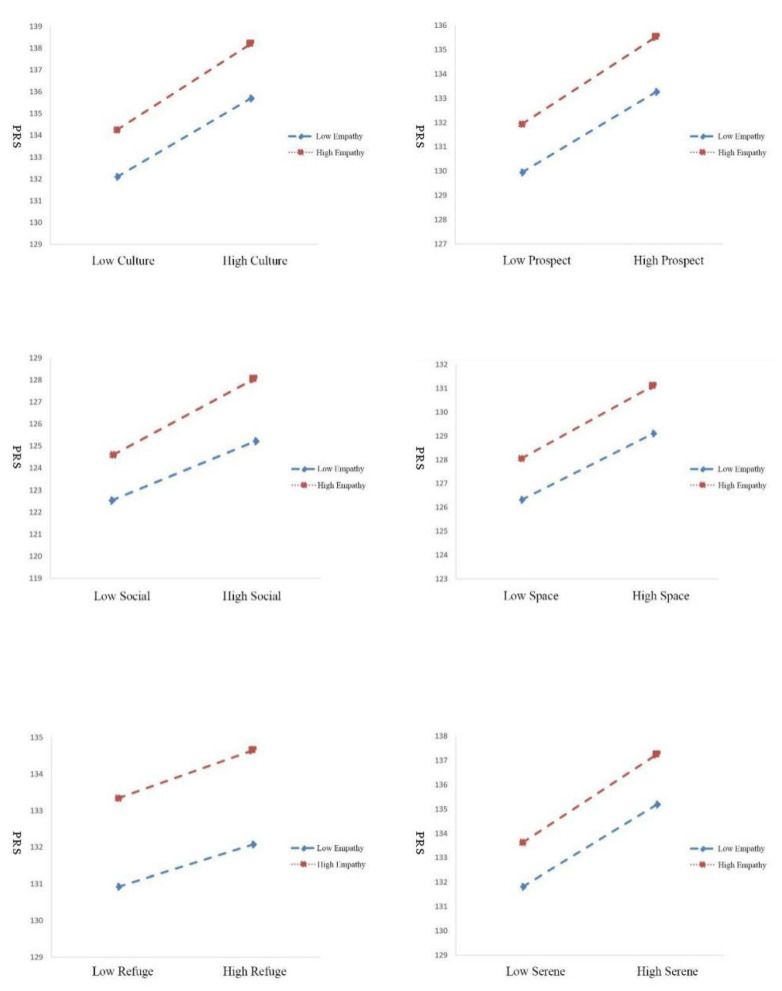
The Moderation Effect of Empathy on PSDs and Attention Restoration.

**Table 1 brainsci-12-00721-t001:** Reliability Test of PSD Scale in Four Parks.

Group	Cronbach’s α	Standardized Cronbach’s α	Items
Group Wangjiang Tower Park	0.951	0.952	35
Group Southern Suburban Park	0.964	0.967	35
Group Huanhuaxi Park	0.969	0.978	35
Group People’s Park	0.944	0.970	35

**Table 2 brainsci-12-00721-t002:** KMO and Bartlett Test of PSD Scale in Four Parks.

Group	KMO Sampling Adequacy	Bartlett’s Test of Sphericity		
		Approx. Chi-Square	df	Sig.
Group Wangjiang Tower Park	0.823	2072.317	595	0.000
Group Southern Suburban Park	0.903	2719.975	595	0.000
Group Huanhuaxi Park	0.891	3652.128	595	0.000
Group People’s Park	0.882	2956.959	595	0.000

**Table 3 brainsci-12-00721-t003:** Multiple Regression: Effect of PSDs on Components of Attention Restoration (N = 348).

	B	Standard Error	Standardized Beta Coefficient	Sig.	R-Squared Variation	F-Test Variation	DW Test
Being Away					0.393	30.286 **	1.890
Culture	0.069	0.149	−0.031	0.642			
Prospect	0.009	0.107	−0.006	0.935			
Social	0.136	0.098	−0.086	0.165			
Space	0.166	0.083	0.158	0.046 *			
Rich in Species	0.131	0.076	0.136	0.087			
Refuge	0.169	0.066	0.235	0.011 *			
Serene	0.663	0.095	0.448	0.000 **			
Fascination					0.404	29.974 **	2.000
Culture	0.290	0.159	0.123	0.068			
Prospect	0.078	0.114	0.047	0.493			
Social	0.014	0.104	0.008	0.894			
Space	0.043	0.089	0.039	0.630			
Rich in Species	0.275	0.081	0.272	0.001 **			
Refuge	0.275	0.070	0.367	0.000 **			
Serene	0.336	0.101	0.217	0.001 **			
Extent					0.131	6.622 **	1.817
Culture	0.468	0.335	0.114	0.163			
Prospect	0.050	0.241	−0.017	0.837			
Social	0.664	0.219	−0.231	0.003 **			
Space	0.130	0.187	−0.068	0.488			
Rich in Species	0.304	0.172	−0.173	0.077			
Refuge	0.385	0.147	−0.295	0.009 **			
Serene	0.306	0.213	0.113	0.151			
Compatibility					0.293	19.573 **	1.892
Culture	0.334	0.179	0.132	0.063			
Prospect	0.240	0.129	0.135	0.064			
Social	0.090	0.117	0.051	0.444			
Space	0.006	0.100	−0.005	0.955			
Rich in Species	0.137	0.092	0.127	0.136			
Refuge	0.130	0.079	0.162	0.099			
Serene	0.299	0.114	0.180	0.009 **			

* *p* < 0.05, ** *p* < 0.01.

**Table 4 brainsci-12-00721-t004:** Hierarchical Regression: Test of Moderation Effect of State Empathy on the Interaction between the PSDs and Attention Restoration.

	PRS (Non-Interaction)		PRS (Interaction)	
Culture	0.159	2.632	0.218	3.434
Empathy	0.451	7.553	0.469	7.896
Interaction			0.140	2.799
R-squared	0.353		0.368 **	
F	26.499		24.632 **	
Prospect	0.281	5.276	0.286	5.438
Empathy	0.390	7.246	0.429	7.830
Interaction			0.132	2.979
R-squared	0.390		0.405 **	
F	31.023		28.883 **	
Social	0.280	5.223	0.253	4.802
Empathy	0.387	7.117	0.477	8.375
Interaction			0.196	4.316
R-squared	0.389		0.421 **	
F	30.904		30.771 **	
Space	0.336	5.307	0.36	5.875
Empathy	0.308	4.844	0.378	5.996
Interaction			0.224	5.022
R-squared	0.390		0.433 **	
F	31.094		32.298 **	
Rich in Species	−0.066	−1.413	−0.087	−1.780
Empathy	0.573	11.959	0.595	11.846
Interaction			0.069	1.446
R-squared	0.344		0.348	
F	25.432		22.586 **	
Refuge	0.254	5.614	0.218	4.698
Empathy	0.459	9.773	0.503	10.315
Interaction			0.133	2.960
R-squared	0.396		0.411 **	
F	31.819		29.572 **	
Serene	0.259	4.042	0.311	4.785
Empathy	0.367	5.720	0.390	6.132
Interaction			0.159	3.351
R-squared	0.370		0.390 **	
F	28.535		27.123 **	

** *p* < 0.01.

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
