# Peer review of "Relationship between PSD of Park Green Space and Attention Restoration in Dense Urban Areas"

_brainsci, 2022, doi:10.3390/brainsci12060721_

Round 1

Reviewer 1 Report

 This paper aims to study the relationship and interaction between the perceived sensory dimensions of urban park green space, and attention restoration, The use of statistics and detailed analysis are commendable. However, the following points are problematic and need to be corrected.

1) Please increase the number of references.

In "1. Introduction," "Attention restoration" is discussed, but there are few references to PSD, and the logic is not structured. In particular, more citations and discussion of the PSD and evaluation methods used by the authors should be provided.

(2) The hypothesis proposed in "1.4. Research Design" needs to be added to the explanatory text.

181-183

"Based on the analysis of the existing relevant literature, the research mainly focuses on the effects of perceived sensory dimensions (PSDs) on attention restoration and the moderation effect of perceived sensory dimensions on attention restoration." 

This sentence alone does not tell us why you created the hypothesis. You should add to it. In particular, please specify "the existing relevant literature.

(3) Please add.

205-208

"The second part is concerned with qualitative analysis of the characteristics of the urban park green space based on PSDs, based on 1000 randomly selected questionnaires, defining the following eight dimensions: Serene, Space, Nature, Rich in Species, Refuge, Culture, Prospect and Social."

Please add how the "1000 randomly selected questionnaires" were collected and analyzed.

4)

223-227

"Finally, 87 validated questionnaires were collected in Group Wangjiang Tower Park, with a 96.67% validation rate, 90 questionnaires in Group Southern Suburban Park, with a 100% validation rate, 85 questionnaires in Group Huanhuaxi Park, with a 94.44% validation rate, and 86 The total number is 348." 

This sentence alone does not give a clear overview of the survey. The sentence needs to be added.

(4-1) First, there must have been multiple candidates in the target cities. The reason why the survey location was chosen from among them should be clearly described.

(4-2) The summary of the survey is unclear.

13-16

"Therefore, we conducted an on-site questionnaire survey in four typical parks in Chengdu and recorded age, sex, daily stress, frequency of visits to parks, and other basic information from the respondents.”

These details should also be included in the paper.

(4-3) The sentence used in the questionnaire should be included in the paper to the extent possible.

(4-4) Basic demographics of the respondents should be included in the paper.

4-5) The number of respondents is less than 100 in each of the four locations. The validity of this should be added.

4-6) Please write an overview of the survey locations. The photos shown are of the surveyed landscape. Since you are examining the impact of the green spaces experienced by respondents, be sure to include photos of the green spaces they are looking at, etc.

4-7) Are the manuscript's results reproducible based on the details given in the methods section? Please review again.

5) After you have added 1) through 4) above, "3. Results" and "4. After you have added the above 1) to 4), please re-consider the contents of "3.

After reading this section, I have the following three questions

Are they easy to interpret and understand? Are the data interpreted appropriately and consistently throughout the manuscript? Are the data interpreted appropriately and consistently throughout the manuscript?

Again, please state clearly, appropriately interpreted and consistently.

6)

269-270

(data is not given in this paper).

 The meaning of this message was unclear. Please add or delete.

7) Please consider how to describe the data.

217-223

"The PRS is significantly reliable and valid, including four components: Being Away, Fascination, Compatibility, and Extent, which can The correlation coefficient of the four subscales and the total scale is between 0.724-0.943, which The Cronbach alpha co-efficient of the total scale and four subscales is between 0.769-0.936, and their split-half reliability is between 0.695 and 0.903."

This statement is found in "2. Materials and Methodology." Is this a result or a method of analysis?

If necessary, please move it to Results or consider how to describe it.

Author Response

Dear Reviewer thank you for your valuable comment our manuscript improves a lot

Reviewer 2 Report

The paper presents a very interesting study on the relationships between the perceived sensory dimensions of parks in dense urban areas and the attention restoration. The study is interesting and quite innovative for the location of the study (parks in areas with more than 15,000 inhabit./sq. km) and for the investigation of the combination of different dimensions.

The research is well structured and the method and tools used are adequate.

The results are encouraging and constitute a good base for future researches. As highlighted by the authors, the study has several limitations, but future studies can confirm these first results.

There are only some suggestions and requests of clarifications that I put in the comments (highlighted in yellow) in the PDF attached.

Author Response

Dear Reviewer thank you for your valuable comments our paper improves a lot. please find the attached word file (author reply) 
